# Historical Appreciation of World Health Organization’s Public Health Paper-34: Principles and Practice of Screening for Disease, by Max Wilson and Gunnar Jungner

**DOI:** 10.3390/ijns11030056

**Published:** 2025-07-21

**Authors:** Peter C. J. I. Schielen

**Affiliations:** International Society for Neonatal Screening, Reigerskamp 273, 3607 HP Maarssen, The Netherlands; peter.schielen@isns-neoscreening.org

**Keywords:** Maxwell Wilson, Gunnar Jungner, screening principles, W.H.O. public health papers monograph 34, Wilson and Jungner principles

## Abstract

Biographies of Max Wilson and Gunnar Jungner were published in 2017 and 2020. An in-depth appreciation of the Wilson and Jungner principles, and the publication they were presented in, ‘Principles and Practice of Screening for Disease’, published as nr. 34 in the Public Health Paper-series of the World Health Organisation (W.H.O), called PHP-34 hereafter, was not published as yet. Here an analysis is given of PHP-34 and the ten screening principles, focusing on three subjects. First, by careful analysis of PHP-34, the literature published in the peer reviewed scientific literature, and other sources, the historical background and origin of the ten principles is determined. Second, the precise composition of PHP-34 is described, as parts of the monograph were derived from other seminal works published between roughly 1950 and 1965. Third, it is determined what the contributions of both authors of the monograph were. Results together are discussed in relation to the time PHP-34 was conceptualized and the importance of PHP-34 and the ten principles in the current era. Results show that in the 15 years preceding the publication of PHP-34, many principles of screening were published by authors in the United States of America, a selection of which ended up in PHP-34. Secondly, about 33% of the 145 pages of PHP-34 are drawn from other publications and studies on screening. Thirdly, the case can be made that the actual writing of PHP-34 was done (almost) entirely by Wilson. Regardless, Wilson and Jungner to this day should be applauded for their work. It is a testimony to the value of PHP-34 that we are still reflecting upon, discussing and seeking to intelligently apply the screening principles almost 60 years after their original publication.

## 1. Introduction

The opening of the 9th International Conference on neonatal screening of the International Society of Neonatal Screening in the Hague, The Netherlands (September 2016), was dedicated to Max Wilson, Gunnar Jungner and their 1968 W.H.O. Monograph ‘Principles and practice of screening for disease’. At the welcome session, the descendants of Wilson and Jungner gave personal accounts of their fathers’ lives. In the process of organizing this session, much information was gathered on the lives of the two authors, as well as their seminal work, ‘Principles and Practice of Screening for Disease’, published as nr. 34 in the ‘Public Health Papers’-series of the World Health Organisation (W.H.O), hereafter called ‘PHP-34’ [1]. Biographies on Jungner [2] and Wilson [3] have meanwhile been published. To complete the trilogy, this study describes the history of the ‘Wilson and Jungner Principles of screening for disease’ and PHP-34.

The Wilson and Jungner principles for screening for disease are universally known among professionals in the field of population screening; this pertains mostly to the ten lines on page 26 and 27 in chapter 2, called ‘Principles’. PHP-34 counts 162 more pages, and it is of interest what comprises these pages. Thus, a quantitative and qualitative analysis of PHP-34 is performed to determine what the original concepts and intellectual property of the authors were and what was inspired by colleagues or drawn from other publications, to support the validity and applicability of the principles. Secondly, the historical background of the ten principles themselves is determined by reviewing the work of predecessors and contemporaries of Wilson and Jungner. Thirdly, it is determined what the contributions of both authors to PHP-34 were.

Thus the aim is to provide a better understanding of the origin and making of PHP-34, against the background of the scientific network of the authors. That understanding is used to explain the importance of the principles to this day.

## 2. Background to Quantitative Analysis of PHP-34

To answer the research questions of this study, PHP-34 was read in its entirety, and the origin of the principles was traced back through studying the literature cited in depth. Where possible, references in the cited literature were read to identify root sources. For the quantitative analysis of PHP-34, per chapter and paragraph, a rough estimate was made of the fraction of a paragraph that could be attributed to either Wilson or Jungner (in terms of the part being the actual intellectual property and the author being the actual writer of a paragraph or section). In a second quantitative analysis, an evaluation was performed of the parts of PHP-34 that could be attributed to cited publications in PHP-34. Fractions in Figure 1 are an estimate of the number of pages per chapter, paragraph or section that are directly drawn from those publications. All estimates were performed by the author–there was no independent review and hence data should be considered as indicative, and sufficient to make a point.

## 3. Quantitative Analysis of PHP-34

### 3.1. A Quantitative Analysis of PHP-34—Introduction to the Chapters

The original edition of PHP-34 counts 163 pages, divided into Title pages (five pages), Contents (one page), Preface (two pages), Definitions (two pages), Introduction (three pages), Principles (26 pages), Practice (38 pages), Illustrative examples (56 pages), Methodological trends in screening (12 pages), Conclusions (6 pages) and References (11 pages). See Figure 1—middle ring—for a donut chart of the relative contributions of the parts to the complete monograph.

The Preface of PHP-34 states that in the era wherein communicable disease in developed countries have become less important ‘as killers’ than chronic disease such as cardiovascular disease and cancer, “Screening for disease will therefore become more important and knowledge of principles and practice should be part of the intellectual equipment of all those concerned with control of disease and maintenance of health.”

In the chapter ‘Introduction’ a number of significant phrases are of note. One is that it is written using first person plural (so referring to two writers). Another is that concerning PHP-34, (a) the authors would have liked to be comprehensive concerning early disease detection but because of the vastness of the subject they simply could not; (b) the examples of screening in PHP-34 were chosen because they appealed to the authors personally and (c) they realize that the subject is controversial and hence they add the frequently used quote: “If anywhere we have appeared dogmatic, we hope this may serve to stimulate discussion, since, in the end, real development depends on an exchange of views.”

The subject of Principles and practice of screening for disease is organized in four chapters: (a) the basic principles of early disease detection, (b) practical considerations, including the application of screening procedures in a number of different disease conditions, and (c) present techniques and possible developments in methodology (the chapters ‘Principles’, ‘Practice’, ‘Illustrative examples’, and ‘Methodological trends in screening’).

PHP-34 also has a small chapter ‘Definitions’. On the definition of screening, it is stated that: “For the purposes of this study the definition of “screening” proposed by the United States multi-sponsored Commission on Chronic Illness (CCI) (…) and accepted by the W.H.O. Regional Committee for Europe [4], is adopted.”

### 3.2. Fraction of PHP-34 Drawn from Other Publications

In PHP-34, tables and text from already published manuscripts, sometimes in adapted form, were reproduced quite extensively (with permission of the authors). In Figure 1 the outer donut diagram gives an estimate of the fraction of PHP-34 that was reproduced from other publications. Parts of PHP-34 are drawn from at least 11 (major series of) works: (1) Chronic illness in the United States Volume I to IV, published by the Commission on Chronic Illness (1957) in total over 1000 pages [5], (2) Thorner, R. M. & Remein, Q. R. (1961) Principles and procedures in the evaluation of screening for disease, United States of America [6], (3) Several publications of the Department of Health, Education, and Welfare, Public Health Service, National Center for Health Statistics, Washington DC, (4) several publications of the Registrar General’s statistical review of England and (5) Many reports of the W.H.O.

The work of the Commission on Chronic Illness is cited 16 times (on a total of 243 citations in PHP-34). Thorner and Remein [6] are cited only twice, but by indication in PHP-34, an important part of chapter two is derived from this publication. Publications of the United States Public Health Service are mentioned 15 times in the main text of PHP-34 and cited 7 times. The work of the Registar is mentioned twice and cited 11 times. W.H.O. reports were cited 18 times. In total, 48 pages, or 33%, of PHP-34 (not including title pages and references-in total 145 pages) are reproduced from the aforementioned publications.

### 3.3. Parts of PHP-34 Attributed to Wilson vs. Jungner

The inner ring of Figure 1 gives an estimate on the parts of PHP-34 that can be attributed to Wilson and to Jungner. Through careful analysis of contents and citations of the entire PHP-34, at best, part of the chapter ‘Methodological trends and screening’ has a clear link to the work of Gunnar Jungner (about 3%) and about 85% can be assigned to Wilson. A more in-depth discussion on the reasoning behind this attribution is given in the Discussion section.

## 4. Origin of the Ten Principles

In Table 1, from right to left, an overview is given of the principles of screening, as presented in historical papers on the subject. The outer right column lists the screening principles as presented in PHP-34, and to the left, principles presented in other publications, partly written by Wilson, partly by other authors, are listed. The numbering of the principles is the numbering in the original papers, and where possible similar principles are sorted based on the order of PHP-34.

The ten principles of screening (pages 26 and 27 of PHP-34) are presented without reference to published scientific papers. Incidentally, on page 17 of PHP-34, three papers are cited of authors Chapman (1949) [7], Mountin (1950) [8] and Smillie (1952) [9] that are stated to deal with the motivation for screening. Based on these papers, five ‘chief points’ are presented as numbered items on page 17 of PHP-34. These five items are also listed in the first column of Table 1. Item 2 of this summary of chief points compares to Principle 2, and item 4 and 5 fit Principle 9.

The publication of Levin in 1955 [10] (not cited in PHP-34) for the first time presented ten principles for screening. Levin reported on the results of a conference entitled ‘Preventive Aspects of Chronic Disease’ organized by the Commission on Chronic Illness, March, 1951. Screening and multiple screening were extensively reviewed and, as reported by Levin, the former commission, chaired by Dr. Lester Breslow (see below), proposed the ten “criteria”, “to be used in determining the suitability of a screening procedure for community use”. While the title of the paper is “Screening for asymptomatic disease (Principles and background)”, the ten items in the paper are presented as “criteria”. While not in the same order, eight out of ten criteria (see second column of Table 1) are similar to the Principles in PHP-34.

A third set of principles similar to those in PHP-34 is presented in a publication called ‘Principles and Procedures In the Evaluation of Screening for Disease’, issued as a Public Health Monograph (No. 67) of the US department of Health, Education and Disease, written by R.M. Thorner and Q.R. Remein [6]. Thorner and Remein called these principles, citing M.S. Blumberg (who published the same principles already in 1957 as, ‘the kinds of things that influence the health value of true and false positives and negatives’. Interestingly, in the same work, Thorner and Remein also mentioned the importance of the availability of suitable tests, an item that is therefore also added in Table 1 (third column).

Four years later, in 1965, Wilson for the first time presented the ten principles in a lecture as part of a colloquium ‘Surveillance and early diagnosis in general practice’, 7 July 1965 at Magdalen College, Oxford, United Kingdom (U.K.). The title of Wilson’s contribution was ‘Some principles of early diagnosis and detection’. In May 1966, the proceedings of this colloquium were published [11]. In those proceedings, the principles, or criteria, or suggested requirements of case finding, as they were also referred to in Wilson’s contribution, are listed for the first time in the order we know of in PHP-34. They are phrased sketchily, in contrast to the more elaborate phrasing in PHP-34. Interestingly, around the same time, another publication of Wilson’s mentions five of the ten principles, but in a different order [12]. Then, finally, three years later, the principles are published in PHP-34 in its final order. Interestingly, a typed draft version is available on the internet stamped May 4, 1967—W.H.O. Library [13]. The principles in that version are worded exactly the same as in PHP-34 with one exception; “Case-finding should be a continuing process and not a “once and for all” project” reads “not a “one-time project” in the May 1967 version.

It is interesting that all authors numbered their principles 1–10, except for Remein and Thorner, citing Blumberg (1957) [14] that numbered them a–f (for convenience translated as 1–6 in Table 1). The use of Western Arabic numbering may be a hint that authors had knowledge of an early version of a list of such principles, possibly published in [10].

Only four principles are covered in all four versions of screening criteria presented in Table 1 (indicated by gray shading). All sets of principles indicate that there should be an accepted treatment. All sets recognize the importance of facilities for treatment and diagnosis. Also, the importance of a suitable screening test is acknowledged by all sets, and finally the importance of the acceptability of the screening test to the population is deemed necessary by all sets of principles. In the list of principles in the paper of Levin [10], this is actually covered in criteria (2) and (8). In these principles of Levin also, the limitations of screening are mentioned. Two principles seem to be recognized by three sets, namely those covering cost of screening, including the cost of the screening procedure and personnel (respectively, item 3 of the Levin principles, item 8 of the Magdalen conference principles and in PHP-34). The other principle is on ‘policy on whom to treat as patients’, which seems to be aligned with Levins ‘Acceptability to the professional groups concerned’. There is almost complete agreement between the principles proposed during the Magdalen conference and the final list of principles in PHP-34. Only principle 7 and 9 contain extended wording that expands the meaning of the principles, introducing the importance of a latent phase in the natural history of disease and a more detailed explanation of what is meant with balance of the cost of screening. At the end of Table 1, principles are listed that were suggested only once by authors.

**Table 1 IJNS-11-00056-t001:** Overview of publications listing screening principles, preceding PHP-34, compared to the principles as finally published in PHP-34. Gray shading indicates that principles published in or before 1957 were adopted in the set of ten principles that were published in PHP-34.

By Chapman [7], Mountin [8] and Smillie [9], Published Between 1949 and 1952, as Cited in PHP-34	Screening for Asymptomatic Disease by Levin-1955 [10] (Referring to Conferences held in 1951).	Principles and Procedures in the Evaluation of Screening for Disease by Thorner and Remein [6], Citing Blumberg. 1957 [14]	Some Principles of Early Diagnosis and Detection by Wilson In: Surveillance and Early Diagnosis in General Practice. Proceedings of Colloquium Held at Magdalen College, Oxford (Wednesday, 7 July 1965) [11]	The Case for General Screening Examinations in Middle Age by Wilson [12]	Monograph PHP-34, by Wilson and Jungner (1968) [1]
			(1) Important problem.	(1) Should be an important public health problem (May be common, e.g., diabetes, or rare, e.g., phenylketonuria)	(1) The condition sought should be an important health problem.
	(9) Amenability to treatment of the disease conditions detected	(1) What is the outlook for a person with the disease? The value of a true positive increases as the patient’s chances for cure or shorter convalescence are improved by early detection. In fact, when no health benefits are attributable to the finding of cases, the advisability of screening is doubtful	(2) Accepted treatment.		(2) There should be an accepted treatment for patients with recognized disease.
(2) Provision for diagnosis, follow-up and treatment is vitally important; without it case-finding must inevitably fall into disrepute	(10) Availability of adequate follow-up, including diagnostic and therapeutic care to all social and economic classes of persons in which disease is detected (lack of facilities for ultimate care should not be considered a bar to a screening program, provided there is a possible willingness on the part of the community to deal with needs revealed by the screening procedures).	(2) What facilities exist for treating cases found? The value of a true positive is reduced if inadequate facilities exist for treating those found.	(3) Facilities for diagnosis and treatment.	(4) Should be treatable when diagnosed	(3) Facilities for diagnosis and treatment should be available.
			(4) Recognizable latent or early symptom stage.	(2) Should have a recognizable presymptomatic or early symptomatic stage	(4) There should be a recognizable latent or early symptomatic stage.
	(6) Extent to which the procedure may be useful in screening for several diseases.	(6) Are healthy individuals being sought? Sometimes screening procedures are conducted to identify healthy rather than sick individuals. This may be the case in selecting people for certain jobs or for the armed services, as well as in screening life insurance applicants. In these cases, false negatives are very costly while false positives may not be.	(5) Suitable test or examination.	(3) A suitable test should be available	(5) There should be a suitable test or examination.
	(7) Compatibility of the tests included in the procedure with the purpose of the screening procedure, i.e., the discovery of disease	Tests used in screening must be relatively simple (…). Screening tests should be fairly sensitive, specific, precise and accurate			
	(2) Acceptability to the individual being screened (8) Feasibility of properly interpreting to the public the need for screening, the results of screening, and the limitations of screening	(4) Who is going to do the diagnostic follow up? False positives burden the diagnostic facilities. If the facilities are adequate and diagnostic study is quite inexpensive, then false positives may be less harmful. False positives also serve to discredit screening	(6) Test acceptable to population.	(5) Screening should be acceptable to the public	(6) The test should be acceptable to the population.
			(7) Natural history adequately understood.		(7) The natural history of the condition, including development from latent to declared disease, should be adequately understood.
	(3) Acceptability to the professional groups concerned		(8) Agreed policy on treatment.		(8) There should be an agreed policy on whom to treat as patients.
(4) There is a danger that multiple screening might lead to the neglect of other aspects of community medical care because of the competing cost and possibly also because a false sense of security might be propagated. (5) The effect of multiple screening needs to be evaluated in terms of reduced morbidity and mortality.	(5) Cost, including accessibility of the population to be screened, time required for the test, physical facilities required, and personnel required.		(9) Cost related to other medical care expenditure.		(9) The cost of case-finding (including diagnosis and treatment of patients diagnosed) should be economically balanced in relation to possible expenditure on medical care as a whole.
			(10) Continuing process		(10) Case-finding should be a continuing process and not a “once and for all” project.
**Principles not directly related to final principles in PHP-34**					
(3) Tests must be validated before they are applied to case-finding; harm may result to public health agencies’ relationships with the public (not to mention the direct harm to the public), and with the medical profession, from large numbers of fruitless referrals for diagnosis. (1) Case-finding by multiple screening is a technique well suited to public health departments, whose role is changing.	(1) Scientific validity of the procedure (4) Productivity of the procedure, i.e., the amount, and social and economic significance, of previously unknown disease likely to be detected	(3) What mental status accompanies knowledge or suspicion of the disease? If suspicion of a disease is accompanied by considerable anxiety that may in turn be debilitating, then demonstration of a true negative could provide valuable reassurance and health benefits (5) What is the likelihood of repeat screening within the community? If it is more likely that repeat screening will be carried on within a short period of time, then false negatives could be extremely detrimental. On the other hand, if screening will be repeated in a short period and the disease is not communicable or rapidly progressing, then false negatives would not be so harmful, since there may be a fair likelihood of uncovering the disease the next time			

## 5. Discussion

The Wilson and Jungner principles today are mostly known and applied as ten lines that together define responsible and ethically acceptable population screening. Less known is that the original work counted 163 pages. This analysis of the history and making of the principles and PHP-34 tries to find out whether the appearance of the principles was a sudden occurrence or that there were preceding events and sources. Also, a quantitative analysis of PHP-34 was performed to identify contributions of other authors that were included in PHP34, to support the authors’ points. Finally, an analysis was performed to determine the relative contributions of both authors. These evaluations were performed to provide a better understanding of the meaning of the principles against the background of public health views at the time of the publication of PHP-34, to also explain their significance in the current era.

### 5.1. On the Origin of the Ten Wilson and Jungner Principles

The data in Table 1 show that PHP-34 was not the first publication to present ten principles for screening. Smillie, Chapman, Mountin, Levin, Blumberg and Thorner and Remein published principles similar to the ones published 15 years later in PHP-34 [6,7,8,9,10].

It is interesting to see that six of ten principles (PHP-34 Principles 2, 3, 5, 6, 8 and 9) date back to these early publications. A full analysis of the papers that dealt with these emerging principles [8,9,10,14] would show that they preluded to even more of the principles that were eventually presented in PHP-34. Here, it suffices to conclude that in PHP-34, all these early principles were arranged for the first time in a more or less logical order, that the most important items were covered, and that a full explanation was given, with many examples to test their applicability, in the chapters ’Practice’ and ‘Ilustrative examples’.

It is of interest to consider principles that were introduced but not adopted in the final version in PHP-34. For instance, Levin as a first principle introduced the scientific validity of the procedure. Levin’s fourth principle was “Productivity of the procedure, i.e., the amount, and social and economic significance, of previously unknown disease likely to be detected”. This very much relates to modern concepts of incidental findings in screening and harms of screening. Principles (3) and (5) from the work of Thorner and Remein also never made it to the ultimate ten PHP-34 principles, but touch upon equally interesting screening principles that are strikingly modern: the anxiety caused by a positive screening result (and hence the value of a true negative screening result) and the importance of mitigating that anxiety. And in relation to that, the relative insignificance of false negatives when screening tests are performed in periodic rounds is of note. Similarly important is that the effect of multiple screening needs to be evaluated in terms of reduced morbidity and mortality. These are all concepts that have their place in modern thinking about screening and sometimes appeared in seminal publications that revisited the principles in PHP-34 [15,16].

### 5.2. Attribution of PHP-34 to Wilson vs. Jungner

In the Results section it is indicated (see also Figure 1) that most of PHP-34, if not the entire monograph, was written by Max Wilson. Neither Wilson nor Jungner stated in interviews or correspondence their individual contributions. Here, the presented attributions in Figure 1 and the Results section are substantiated. In Table 2 a timeline is given to support the line of reasoning below.

Wilson must have been familiar with all published sets of principles for screening, meant to protect the population from screening’s harms. Perhaps the very first introduction in Europe of the developments of multiphasic screening in the U.S.A. may have been the 1957 symposium on ‘Public Health Aspects of Chronic Disease’ in Amsterdam, the Netherlands, sponsored by the Regional Office for Europe of the W.H.O. in collaboration with the government of the Netherlands. George Godber, a senior colleague of Wilson was present at that meeting and a paper of Dr. Lester Breslow, ‘Early detection of asymptomatic disease’ was presented [3]. In 1962 Wilson made a visit to the U.S.A on a W.H.O. traveling fellowship with the intent to learn more about this exclusively American concept of early disease detection or (multiphasic) screening. According to a travel itinerary (Appendix A), Wilson visited Dr Quentin Remein on April 20, Dr Morton Levin on April 30, Dr Lester Breslow, Dr. Morris Collen on May 14, and Dr. J.M. Chapman on May 22 (the entire voyage was from April 7 to May 28 and Wilson intended to visit a total of 118 persons). He held discussions with 97 colleagues.

In his article in Lancet in 1963 [17] and a report to W.H.O. in 1962 [24], Wilson presented a prelude to PHP-34. It contained an overview of the concept of screening as perceived in the U.S.A., indicated some of the important problems of screening (e.g., lack of unbiased evaluation, natural history of the disease, effect of treatment), and referred to evaluation projects in the U.S.A., that would later, more elaborated, appear in PHP-34 (e.g., the Oxford Massachusetts, Framingham, Tecumseh, Birmingham, Bedford and San Francisco studies), and also some epidemiological studies performed in the U.K.

Wilson wanted to report on “…what is now being thought and done about screening in the early detection of disease.” The article was meant “to draw some conclusions on ‘multiple screening’ and their possible application in the U.K.”

He presented the merits of population screening but took good note of the critical appraisal of, e.g., Smillie and Mountin, who mentioned the necessity of sound epidemiological studies to avoid harm by indiscriminate screening and the importance of knowing the natural history of the screened disease. And also that without effective treatment, only harm will be done by bringing the condition to the patient’s attention. Smillie was particularly critical on the dangers of screening for disease as a new public health facility. In his words, “the individuals in a community in whom we are interested are not so much the 40 of the 1000 who were selected by these various tests as presumably ill, but the 960 persons who were told that they were presumably well”. Wilson sharply identified one fundamental concern of screening as a public health procedure, namely that responsibility for the participants is incurred by the authority who organized the program. This differed from the classical GP–doctor relationship, although that relationship comes into play when a participant becomes a positive case in a screening program.

To Wilson, authors like Smillie and Mountin for the first time warned of the downsides of health screening, and, to mitigate those risks, they presented rules, requirements, criteria or principles as a benchmark for medical–ethically justified screening. They cautioned for overly optimistic expectations of the relatively new concept of population screening and to protect the population from the disadvantages and dangers of such screening.

In 1964 the W.H.O. meeting in Prague was held (see Table 2) and just a year later, the seminal meeting in Magdalen College took place. In his contribution at the Magdalen College colloquium [11], Wilson elaborated on his Lancet paper. He introduced ten principles of screening and their meaning and importance. It is a lengthy article that is almost entirely committed to stressing the importance of the screening principles. Also, a few examples of disorders that are fit for screening are given, but not as extensive as in PHP-34. In the same Magdalen College Colloquium Proceedings, Jungner had three long and significant contributions (‘Chemical health screening’, ‘Appendix 1: Chemical Health screening’ and ‘Appendix 2: The Health screening program in Värmland’, in total ten pages). The Värmland study was a study that intended to collect chemical analysis data of a cohort of 100.000 inhabitants of the Swedish county of Värmland. It would have been easy to include a comprehensive outline of these studies as examples in PHP-34, similar to the numerous other studies included, but, curiously, these studies were hardly mentioned. PHP-34 was possibly written between July 1965 (Magdalen college colloquium) and May 1967 (first complete draft in W.H.O. library—Table 2). Both Wilson and Jungner published papers during that period. Wilson wrote the previously mentioned paper ‘The case for general screening examinations in middle age’ [12], introducing five of the ten principles that earlier appeared in the Magdalen College procedures, together with elaborate examples of screening efforts, for, interestingly, phenylketonuria and diabetes, extensively discussing the so-called Bedford study, and glaucoma, discussing the so-called Rhondda study. Both the Rhondda and Bedford study will later appear in PHP-34. Given that five of ten principles of the Magdalen proceedings [12] were published in 1966, it could be that the conception of the 1966 paper lay before Wilson’s Magdalen college lecture, where he meanwhile extended his ideas to ten principles (see Table 2).

As discussed previously, it was an assignment to Wilson to go to the United States and visit 97 professionals in the field of screening for disease to disseminate that knowledge in Europe. Thirty three percent of all pages of PHP-34 are reproduced or adapted from other publications, but especially from American authors, and Wilson visited all these authors during his trip in 1962. We can assume that at least these pages ended up in PHP-34 through Wilson.

Of course, Jungner had his own connections to American colleagues, specifically Dr Morris Collen of Kaiser Permanente group [2]. However, the line of their mutual work (laboratory automation) covers only a small proportion of PHP-34. One would expect that especially chapter 5, ‘Methodological trends in screening’, and the paragraph ‘Automatic data handling’, the first paragraph of the chapter ‘Practice’, would refer to Jungner’s experience on those subjects. Notably, Jungners work is not cited here. Only two paragraphs, ‘Clinical methods and laboratory tests used for screening’ and ‘Automation’ in the chapter ‘Conclusions’, refer to two publications of Jungner, in total four pages. Wilson may have written these parts as well, as he was well aware of Jungner’s work on the Värmland studies and the so-called ‘Autochemist’ for automated laboratory analysis. Jungner presented on both subjects at the Magdalen college colloquium, but let us assume that Jungner had some input in this part of PHP-34.

Between July 1965 and May 1967, Jungner gave lectures at a meeting in Elsinore (Denmark) in 1966 on automated data processing in hospitals. His work, together with his brother Ingmar, concentrated on the Värmland project and the development of laboratory automation machines, specifically the so-called ‘Autochemist’ [25]. While Jungner also gave a lecture entitled ‘Health Screening’, at the meeting in Elsinore, work of Jungner focused on technical, analytical and automation aspects of screening, mostly referring to this as ‘Chemical health screening’ [26]. Contrary to Wilson, no publications on the medical–ethical and organizational aspects of screening from Jungner are known.

Additional indications that Wilson was the main, if not only, author of PHP-34 can be derived from an extensive review, entitled ‘Principles and practice of screening for disease. Introducing a new book’ [18]. This was a ten-page interview with Wilson on the occasion of the publication of PHP-34 in the W.H.O. Chronicle, issue 22, on PHP-34, sharing intentions, graphs and figures, as well as the principles itself. This interview did not mention Jungner or his contribution to PHP-34.

In 1968, 1971, 1973, 1986 and even 1994, Wilson continued to publish on the principles of screening, in line with PHP-34, cautioning that screening does harm, the capacity of the network of GPs can easily be overstretched by screening, and also explaining the importance of the principles for screening [19,20,21,22,23,27]. Jungner, who died in 1982, has some publications to his name after 1968, but mostly in Swedish and not on the subjects of PHP-34, but more on automation in laboratory analysis. In an overview of Jungner’s work [26], PHP-34 was only mentioned as a sidenote.

It could be that based on Jungner’s contribution to a W.H.O. conference in 1964 in Prague and Wilson’s contribution on the W.H.O conference in 1965 in Oslo, Wilson and Jungner received the joint assignment to draft PHP-34 by Dr. F. Grundy (assistant Director general of W.H.O. at the time–see [13]). The 1965 Magdalen conference was evidently the moment that Wilson and Jungner were together and could have laid the groundwork for PHP-34. In the absence of modern communication means to work together on a draft, it is likely that it was agreed that Wilson wrote it and that the mutual authorship was based on courtesy; they accepted the assignment together and they were befriended [3].

## 6. Conclusions

The ten principles for ethically responsible screening in PHP-34 were heavily inspired by especially U.S.A. scientists that worked with similar published concepts predating PHP-34 by at least 15 years. Some of the early progenitors of those principles have not made it to PHP-34 but are equally important and maybe should have been added. Additionally, PHP-34 relied for illustrative examples of screening in practice on published studies especially from the U.S.A. used with permission of the authors, making up a considerable part of PHP-34. 

It is important to stress that the principles were directed toward General Practitioners and the society, i.e., those working within primary care and the community. They were designed to work in utilitarian contexts, not clinical ones. It is important to remember this in an era where medical specialists and fast-developing genetic technology determine the debate on screening. It stresses the importance of principles on which to build safely for the entire society and further define those principles by setting criteria and measures to be a decision tool.

We may also conclude that Max Wilson has probably written the entire Monograph. What is far more important though is that Max Wilson and Gunnar Jungner were part of an international community and with W.H.O as a catalyst, they together introduced concepts of screening from the U.S.A. to Europe. Also, the principles were presented in such a way that almost 60 years after their original publication we are still reflecting upon, discussing and seeking to intelligently apply them. We would do well to preserve that rich legacy as a guide for the future.

## Figures and Tables

**Figure 1 IJNS-11-00056-f001:**
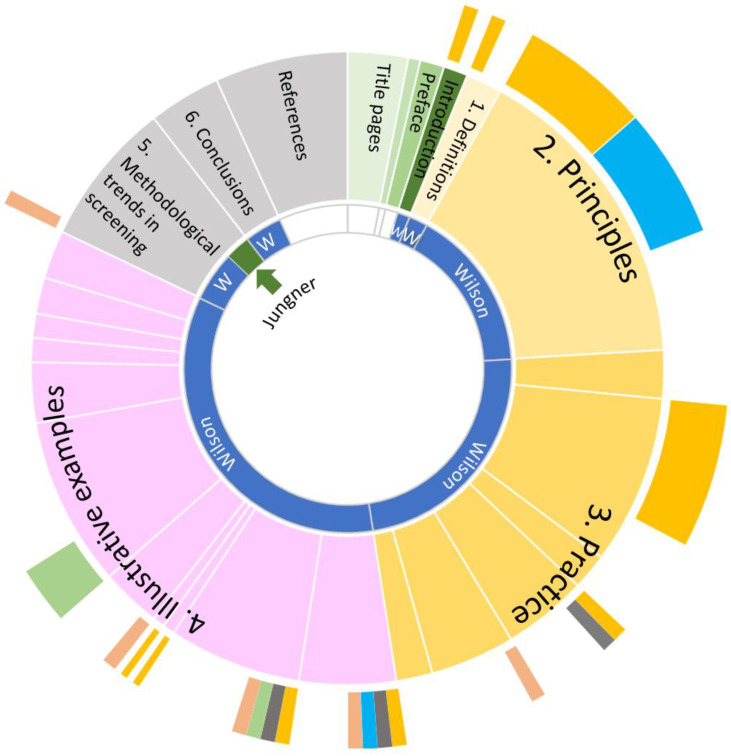
Donut chart of the relative contribution of the various chapters of PHP-34. Inner ring: putative contributions Wilson (Wilson, W, blue) and Jungner (green). Middle ring: chapters and paragraphs (separated by lines) of PHP-34. Outer ring: fractions of chapters and paragraphs traceable to other publications. Yellow: Commission on Chronic Illness (1957); gray: Department of Health, Education, and Welfare, Public Health Service, National Center for Health Statistics, Washington; blue: Thorner and Remein; green: Registrar General’s statistical review of England; orange: W.H.O.

**Table 2 IJNS-11-00056-t002:** Historical overview of papers and events elucidating contributions of Wilson and Jungner to PHP-34.

Year (Month)	Event	Significance
1949	Publication paper Chapman [7]	Published on the concept of multiple screening with some references to ‘Principles’.
1950	Publication paper Mountin [8]	Published on the subject of multiple screening with special reference to test (acceptability to the public, specificity and sensitivity and cost-effectiveness)
1951	Meetings of the Commission on chronic illness	A series of meetings, the results of which ended up in a seminal report; the 1957 publication of Chronic Illness.
1952	Publication paper Smillie [9]	Published on multiple screening with a quite critical view to the advantages of multiple screening.
1955	Publication paper Levin [10]	Published a paper containing for the first time ten principles of screening.
1957	Symposium ‘Public Health Aspects of Chronic Disease’ Amsterdam, the Netherlands sponsored by W.H.O. Europe	Presentation of a paper of Dr. Lester Breslow, ‘Early detection of asymptomatic disease’ before a European audience, including Dr George Godber, senior colleague to Wilson.
1957	Publication ‘Chronic Illness in the United States’, by the Commission on Chronic Illness [5]	Publication of a four-volume work (of which many parts are adopted into the PHP-34) under the editorial supervision of Breslow. Wilson visited Breslow in the U.S.A.
1961	Publication of principles and procedures in the evaluation of screening for disease [6]	A seminal work by Thorner and Remein. Larger parts of this work find their way to PHP-34 and Wilson spoke with Thorner during his travels to the U.S.A.
1962	Travel Wilson to U.S.A.	Wilson visits 97 colleagues in the U.S.A. on a W.H.O. grant, encouraged by Godber. Many of those later receive prominent attention in the Monograph, their work is cited abundantly, and larger parts of the work are adopted in the PHP-34.
1963	Publication paper Lancet Wilson [17]	Report of Wilson of his W.H.O.-funded visit to the U.S.A. In this paper, no clear references to any principles could be discerned yet. Rather, Wilson highlighted the difference between the U.S.A. and U.K. public health landscape and how this will affect the adoption of screening for disease.
1964	W.H.O. meeting Prague	Jungner presenting together with Maria Missir on Public health screening. The name of Maria Missir will not reappear in W.H.O. discussions on screening. In the minutes of this conference, it was stated that all the participants had a keen interest in studying and developing organized screening services.
1965 (July)	Magdelen College Colloquium, Oxford, U.K.	A conference in the U.K. with mostly U.K. representatives, amongst which was Wilson, but also Jungner and Dr. Collen, an American representative presenting on multiphasic screening.
1965 (November)	W.H.O. meeting in Oslo, Norway, with Dr. A. Grundy and Wilson.	It took this second W.H.O. meeting after the one in Prague to start writing the monograph. Wilson gave a report to the W.H.O. Conference ‘Early Detection of Cancer’, Oslo, Norway, 15–19 November 1965.
1966	Publication of J Path Practice paper Wilson [12]	Introduction of 5 of the 10 principles that were published in 1968 in the monograph. Note: this paper could have been conceived before July 1965, so predating the Magdalen College Colloquium.
1966 (April–May)	Elsinore conference	Conference in Denmark where Jungner presented on public health screening and laboratory test-automation
1967 (May)	First draft-Principles and Practice screening for disease [13]	The fact that a full draft was available in May 1967 limits the production time of the Monograph to roughly between December 1965 and April–May 1967
1968	Publication Monograph [1]	
1968	Publication in Chronicles W.H.O. [18]	The report of a 10-page interview with Max Wilson, sharing significant parts of the Monograph. While Jungner is mentioned in the introduction, the interview is with Wilson and is suggestive in many instances that Wilson is the writer of the monograph.
1970 and 1971	Publication papers Wilson [19,20]	On ‘Problems in the evaluation of screening for disease’ and on ‘Principles of screening for disease’
1973	Publication papers Wilson [21]	On ‘Current trends and problems in health screening’.
1982	Jungner dies	-
1986 and 1994	Publication papers Wilson [22,23]	On Screening for visual defects in preschool children with the outcome that more research is needed to justify screening and a retrospective on medical screening, from beginnings to benefits.
2009	Wilson dies	

## Data Availability

No new data were created for this publication.

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
