# Peer review of "Historical Appreciation of World Health Organization’s Public Health Paper-34: Principles and Practice of Screening for Disease, by Max Wilson and Gunnar Jungner"

_2409-515X, 2025, doi:10.3390/ijns11030056_

Round 1

Reviewer 1 Report

Comments and Suggestions for Authors

Thanks for the article to let us know that the Wilson and Jungner principles today are not only ten lines but in an original work counted 163 pages, and the principles were not a sudden occurrence. I have not additional comment, but just feel difficulties in reading Table 1. I understand that the author wants to put the publications in chronological order, it may be easier to read if the 1968 10 principles are put on the first column.

Author Response

Dear Madam/Sir,

Thank you for reviewing the manuscript and your comments on the readability of table 1. I agree, it is by its nature a complex and not a very palatable table and I have tested your valued suggestion to start with the final column and work from there. We read tables from left to right and chronologies from past to present at that element gets lost when starting with the final column. In practice, disturbing that element made the end-result not an improvement.

Still, thank you again for your suggestion, that I valued and seriously considered and I hope it is acceptable for you that I did not adopt this suggestion at this time. 

Sincerely Yours

Peter Schielen

Reviewer 2 Report

Comments and Suggestions for Authors

Thank you for the opportunity to review this manuscript, covering the history of a document and concept many in newborn screening refer to frequently, but which most of us have probably not read.  Nor indeed, did we likely appreciate the full depth of the manuscript which originated the "Wilson and Jungner criteria".  Like most, I knew of the 10 point list, and not much else.  This is an in-depth look at where it came from and how it originated.  

I have a couple minor comments about the manuscript, but nothing major content wise.  I think it is important to understand that the key concepts we rely on are often a part of a larger work, and we may be missing something by only focusing on the highlights.

Intro:  try to avoid one sentence paragraphs; there are several that could be combined

Figure 1:  Reformat the image to remove the red squiggly “misspelling” line from under “Jungner” in the innermost circle.  Also, if this is meant to refer only to the green segment, consider some sort of key or arrow for an indicator.

Throughout:  there’s a few places where the plural (Wilsons and other names) is used in place of the possessive (Wilson’s).  Suggest a search to find and identify all of these.

Line 280:  “…learn more about this exclusively USA concept..”  would switch to the adjective “American” or rephrase the sentence as "USA" is not an adjective.

Author Response

Dear Madam/Sir,

Thank you for your thorough review and comments.

Please find a point-by-point rebuttal of your comments below.

1) Try to avoid one-sentence paragraphs.

I have joined paragraphs to overcome this.
See changes in lines 49, 51 and 98.

The one sentence paragraph in line 209-211 was expanded with one sentence to clarify the significance of the observation in that paragraph.
See changes in line 212-213.

2) Repair items in Figure 1.
On the red underlining-thank you, well spotted and corrected.

On the indication of the meaning of the green segment:
Thank you for this thoughtful suggestion. I followed your suggestion to use an arrow to indicate the meaning of the green segment also removing the question mark behind Jungners name.
See changes in Figure 1.

3) On the use of Wilsons vs. Wilson’s.
Thank you again-well spotted and I repaired that throughout the document.
See changes in line 197, 201, 337, 379.

4) Thank you-I changed USA to American
See change in line 282.

Reviewer 3 Report

Comments and Suggestions for Authors

The paper is structured as research, there is not evidence of research. While the topic of screening is always of interest to the readers of this journal, the author fails to provide historical context for his paper. Nor does the author analyze or provide an in-depth review of the contributions to screening literature by contemporary scientists. The Table and its references provided are of value, but do not constitute research, per se.

Comments on the Quality of English Language

Many "run-on" sentences. A few are incomplete

Author Response

Dear Madam/Sir,

Thank you for your critical review of this manuscript and thank you for the recognition that the paper is relevant to the authors of International Journal of Neonatal Screening.

See below for a coverage of your critique.

1) On the point of the structure.

I agree with you that this is not a research paper, with a historical context leading to a research question, a set of experimental methods and a results section with a collection of results, and results discussed in relation to the work of contemporary authors in the same field. 

I have now removed the headings 'Methods' and 'Results' and changed them to headings that are more narrative, to steer away from the standard structure of a full scientific paper. Thus, the structure of the paper better fits its aim, a historical appraisal of screening principles that have been central to screening, including neonatal screening, in the past 60 years. I hope this covers your critique. 
See changes line 56 and 70

Round 2

Reviewer 3 Report

Comments and Suggestions for Authors

no comments